# Structural Insights into Lactococcal Siphophage p2 Baseplate Activation Mechanism

**DOI:** 10.3390/v12080878

**Published:** 2020-08-11

**Authors:** Silvia Spinelli, Denise Tremblay, Sylvain Moineau, Christian Cambillau, Adeline Goulet

**Affiliations:** 1Architecture et Fonction des Macromolécules Biologiques, Aix-Marseille Université, Campus de Luminy, 13288 Marseille CEDEX 09, France; silvia.spinelli@afmb.univ-mrs.fr; 2Architecture et Fonction des Macromolécules Biologiques, Centre National de la Recherche Scientifique (CNRS), Campus de Luminy, 13288 Marseille CEDEX 09, France; 3Département de Biochimie, de Microbiologie, et de Bio-Informatique, Faculté des Sciences et de Génie, Université Laval, Québec, QC G1V 0A6, Canada; Denise.Tremblay@greb.ulaval.ca (D.T.); Sylvain.Moineau@bcm.ulaval.ca (S.M.); 4Groupe de Recherche en Écologie Buccale, Faculté de Médecine Dentaire, Université Laval, Québec, QC G1V 0A6, Canada; 5Félix d’Hérelle Reference Center for Bacterial Viruses, Faculté de Médecine Dentaire, Université Laval, Québec, QC G1V 0A6, Canada

**Keywords:** *bacteriophages*, *Lactococcus lactis*, *Siphoviridae*, nanobody, electron microscopy, infection mechanism

## Abstract

Virulent phages infecting *L. lactis*, an industry-relevant bacterium, pose a significant risk to the quality of the fermented milk products. Phages of the Skunavirus genus are by far the most isolated lactococcal phages in the cheese environments and phage p2 is the model siphophage for this viral genus. The baseplate of phage p2, which is used to recognize its host, was previously shown to display two conformations by X-ray crystallography, a rested state and an activated state ready to bind to the host. The baseplate became only activated and opened in the presence of Ca^2+^. However, such an activated state was not previously observed in the virion. Here, using nanobodies binding to the baseplate, we report on the negative staining electron microscopy structure of the activated form of the baseplate directly observed in the p2 virion, that is compatible with the activated baseplate crystal structure. Analyses of this new structure also established the presence of a second distal tail (Dit) hexamer as a component of the baseplate, the topology of which differs largely from the first one. We also observed an uncoupling between the baseplate activation and the tail tip protein (Tal) opening, suggesting an infection mechanism more complex than previously expected.

## 1. Introduction

Over a century after their discovery, the vast majority of known bacterial viruses (bacteriophages or phages) are still poorly understood [1]. Only a few model phages, mostly infecting *Escherichia coli* strains, have been thoroughly characterized. Studies on phages infecting Gram-positive bacteria have lagged behind but have been steadily increasing in the past decade. Among others, virulent phages infecting the economically-important lactic acid bacteria (LAB) have been progressively more characterized due to their negative impact on food fermentation processes [2].

*Lactococcus lactis* is the most widely used lactic acid bacteria globally as it is needed to manufacture several cheeses. Numerous lytic phages infecting several strains of this bacterial species have been isolated and characterized worldwide. Such phage monitoring program is essential to adapt antiviral strategies and to avoid their damaging effects during cheese production [3]. These lactococcal phages have also been divided into several groups (e.g., Skunavirus, P335, etc.) based on morphological and genomic approaches [4]. Virulent phages belonging to the newly created Skunavirus genus (previously 936) are by far the most predominant in dairy factories and they currently belong to the *Siphoviridae* family (double-stranded DNA genome and non-contractile tail) of the Caudovirales order. Phage p2 is the model phage for this viral genus and it infects *L. lactis* MG1363, a laboratory strain used in many studies on Gram-positive bacteria [5]. This phage has a genome size of 27,595 bp and 50 annotated open reading frames. Of interest here, phage p2 uses a multi-protein baseplate at the tip of its tail to bind to its cell wall polysaccharide (CWPS) receptor at the host cell surface [6,7,8].

The structure of the phage p2 virion was previously determined by negative-staining electron microscopy (nsEM) [9]. The baseplate sub-structure at 21 Å resolution appears as rather compact at the tail distal end. The components of the baseplate, (ORF15, ORF16, and ORF18) were recombinantly produced and assembled into a 1.2 MDa complex [10], which was subjected to structural analysis [6]. Firstly, crystallization assays of the purified baseplate complex resulted in poorly diffracting crystals, likely due to the fast disassembly of the complex and its structural heterogeneity. As we previously showed that a llama immunoglobin VHH domain, hereafter referred to as VHH5, was able to complex the receptor-binding protein (RBP, ORF18) and neutralize p2 infection [11], we mixed our purified baseplate complex with an excess of VHH5 and obtained crystals diffracting to 2.6 Å resolution. Thus, we were able to build an atomic model of the baseplate containing the distal tail protein (Dit, ORF15), the tail associated lysin protein (Tal, ORF16), and the RBP complexed by VHH5 binders [6]. We also determined the cryo-electron microscopy (cryoEM) structure of a freshly purified and glutaraldehyde cross-linked baseplate at 26 Å resolution [6]. The baseplate architectures in the cross-linked complex and in the native p2 virions were similar within their resolution limit (Figure 1A,B). Furthermore, the overall architecture of the baseplate crystal structure, depleted of the VHH5s, was in agreement with the EM structures (Figure 1C–E). However, the baseplate atomic model containing six Dit, three Tal and six trimers of RBP corresponded only to a part of the cross-linked baseplate EM structure, as a two ring-shaped density could not be immediately assigned to any component of the structure (Figure 1D,E). Since the recombinantly produced baseplate contained only Dit, Tal, and RBP, these densities had to be fulfilled by one of the three components, and a second Dit hexamer was the most likely candidate. Therefore, we proposed that one of the rings could be formed by a Dit N-terminal domain back to back with the first Dit N-terminal domain (referred to as Dit-1) hexameric ring, while the second ring (referred to as Dit-2) could be formed by the Dit C-terminal galectin domains, rearranged to form a hexamer [6]. We noticed that small densities point out of Dit-2 galectin ring and contact the RBPs, probably keeping the baseplate in a resting state.

In the baseplate crystal structure, the RBP orientation was somewhat surprising since the receptor-binding domains were found to point towards the virion capsid (Figure 1C) and not in the opposite direction, as intuitively thought to interact with the host cell wall receptor. However, when Ca^2+^ (or Sr^2+^) was added to the purified baseplate complex, the baseplate exhibited a different conformation in which the receptor-binding domains point downwards in the direction opposite to the capsid (Figure 1F). In addition, the Tal trimer adopted an open conformation instead of a closed one in the purified baseplate-VHH5 complex [6]. 

In both baseplate conformations, an extension of the Dit galectin domain, the “arm”, holds the RBPs by inserting its “hand” within the trimeric RBP “shoulder” domain (Figure 1C,F) [6]. In this context, it was proposed that the structure obtained in the presence of Ca^2+^ was an actived form of the phage baseplate [6]. It was also suggested that the baseplate transition from resting to activated state could be the trigger for Tal opening and subsequent release of the tail tape measure protein (TMP), followed by DNA ejection. However, we were unable to obtain this baseplate activated state in p2 virions, despite many attempts, which included the addition of divalent cations (Ca^2+^, Sr^2+^), hampering somewhat the validity of the suggested activation mechanism.

Here, we present further structural analyzes of the phage p2 baseplate, proposing a topological model of the baseplate resting state, and interpreting a new nsEM structure of the virion, in its active conformation. These novel views on the activation mechanism of phage p2 may be applicable to other members of the *Siphoviridae* family exhibiting an activation mechanism [7,12].

## 2. Materials and Methods

### 2.1. Phage Production and Purification

Phage p2 and its host *L. lactis* MG1363 were obtained from the Félix d’Hérelle Reference Center for Bacterial Viruses (www.phage.ulaval.ca). *L. lactis* MG1363. For phage amplification, CaCl_2_ was added to the medium at a final concentration of 10 mM. Phage p2 was amplified and purified as reported previously [9]. Purified phages were conserved at 4 °C in buffer containing 50 mM Tris-HCl pH 7.5, 100 mM NaCl, 8 mM MgSO_4_.

### 2.2. Negative-Staining Electron Microscopy Sample Preparation

A 5-μl droplet of p2 virions incubated overnight with molar excess of VHH5 was applied onto a glow-discharged 300-mesh copper carbon-coated grid (Agar Scientific, Oxford Instruments, Gometz la Ville, France) for 1 min. The excess of sample was removed by side-blotting and the grid was washed with a 10-μL droplet of water. Then, the grid was stained with 1% (*w*/*v*) uranyl acetate solution for 20 s and blotted dry.

### 2.3. Negative-Staining Electron Microscopy Data Collection and Image Processing

The sample was imaged on a T12 Spirit transmission electron microscope operating at 120 kV. A total of 261 images were collected with a pixel size of 3.46 Å/pixel at the specimen level. Images were acquired using a Veleta CCD camera (Olympus, München, Germany) with an electron dose of −20 e^−^/Å^2^. Micrographs were imported in RELION-3.0 for all subsequent image-processing tasks [13]. Estimation of the Contrast Transfer Function was performed with CTFFIND4 [14] and particles were manually picked. A total of 293 particles were extracted, normalized, and subjected to rounds of reference-free 2D classification in RELION-3.0 (to remove bad particles). An initial 3D reference was generated using a Stochastic Gradient Descent algorithm in RELION-3.0 and subsequent rounds of 3D refinement were performed imposing D6 symmetry with 261 particles. Particles selected after 2D classification were also subjected to 3D classification to identify potential structural heterogeneity. A total of 68 particles (26% of total particles) and 193 particles (74% of total particles) contributed to two 3D classes, Class 1 and Class 2, and were individually subjected to rounds of 3D refinement imposing D3 symmetry (Appendix A). The resolution of the three final 3D reconstructions was calculated from Fourier shell correlations at 0.143 (Appendix A).

### 2.4. 3D Structure Modeling of Dit-2

Modeling of Dit-2 3D structure was performed with Coot [15,16]. The Dit-2 galectin domains were positioned within the EM map of the baseplate rest form (EMD-1699) [6] above the ring of Dit-2 N-terminal domains. The geometry of the linkers between the N-terminal and C-terminal domains was regularized using the “regularize zone” Coot option. The structure of the arm-hand extension of the galectin domain was modelled within the EM map density (EMD-1699) bridging the galectin domain to the RBP head, and its geometry was regularized.

### 2.5. Structure Analyses and Representation

Rigid-body fitting of baseplate and baseplate component crystal structures within electron microscopy 3D reconstructions was performed using UCSF Chimera X [17]. The quality of the fit was assessed by cross-correlation between the experimental and simulated maps. The final baseplate topological models fitted within the EM 3D reconstructions are available as Cα models. The difference map between the 3D reconstructions of the activated baseplate with Tal in closed and open conformation was calculated using TEMPy [18] in CCP-EM [19]. The EM maps were low-pass filtered at the same resolution and aligned, and the negative voxels in the subtracted map were set to 0. UCSF Chimera X [17] was used to make figures.

### 2.6. Data Availability

The negative-staining EM maps and the molecular models have been deposited in the Electron Microscopy Data Bank and RCSB Protein Data Bank under the accession codes EMD-11226 and PDB ID 6ZIH (p2 virion baseplate bound to VHH5), EMD-11225 and PDB ID 6ZIG (p2 virion baseplate bound to VHH5, 3D class with closed Tal trimer), EMD-11224 ((p2 virion baseplate bound to VHH5, 3D class with open Tal trimer), and PDB ID 6ZJJ (topological model of p2 virion baseplate in resting conformation, related to EMD-1699).

## 3. Results

### 3.1. Topological Model of the p2 Virion Baseplate in Its Resting State

As indicated above, the baseplate architectures in phage p2 virion, in the recombinantly produced cross-linked complex and in the crystal structure with bound VHH5s, are similar: they all exhibit the receptor-binding domains pointing towards the capsid, and the Tal trimer is closed (Figure 1A–C). With the aim of getting a complete topological model of the p2 baseplate in its resting state, we have rigid-body fitted the baseplate crystal structure (PDB ID 2WZP) [6] into the distal tail end of the p2 virion nsEM structure (Figure 1D). Although the overall fit was of good quality, we noticed that the RBPs leaned towards the baseplate central axis, an effect probably due to the VHH5s in the crystal structure (Figure 1D). Therefore, we have rigid-body fitted individually each RBP trimer into the map. The resulting Dit-Tal-RBP assembly gave an excellent fit (Appendix A) (Figure 2A). 

Then, we positioned a second Dit ring back to back with the first one, using the Dit-Dit back to back assembly observed between symmetry-related molecules in the crystal packing of the activated baseplates (PDB ID 2X53) [6]. This resulted in a nice fit of the Dit-2 N-terminal ring, while the galectin domains were out of the map. Therefore, we fitted the Dit-2 galectin domains, as a single rigid body, into the second map ring that contacts the Major Tail Protein (MTP) hexamer, and we modeled the backbone of the linkers connecting the Dit-2 N-terminal and galectin domains using the coot software [15,16]. In this new position of the Dit-2 galectin domains, the tips of the arm-hand extensions were out of the map. Therefore, we also modeled their backbone to fit them into the map, although their exact position could not be ascertained. Our final Dit-2 model led to a very good fit in the nsEM map (Appendix A) (Figure 2B). The six Dit-2 galectin domains contact each other and form a continuous ring (Figure 2C).

Our topological model including Dit-1, Dit-2, Tal, and the RBPs, reveals structural variations between the two Dits enabling the assembly of the baseplate in its resting state. In particular, the Dit-2 galectin domains have rotated towards the tail axis by approx. 50 degrees around the pivot amino acid residue 133, and the stretch of amino acid residues connecting the N-terminal and galectin domains (residues 133–139) has extended itself in the direction of the capsid (Figure 3A,B). Moreover, the Dit-2 arm-hand extension is bent towards the baseplate central axis, instead of being projected radially as in the Dit-1 ring (Figure 3). In such conformation, the Dit-2 arm-hand extension (residues 145–186) contacts one receptor-binding domain in a crest of the β-barrel (residues 195–198 and 204–206), a position that is close, but distinct, to the saccharidic receptor-binding site located at the interface between two β-barrels of the trimeric RBP (Figure 2C).

### 3.2. Topological Model of the Virion Baseplate in Its Activated State

The neutralizing VHH5s bind at the interface between RBP β-barrels, thereby preventing host receptor binding [6] (Figure 2C). Therefore, our topological model of the baseplate resting state suggests that binding of VHH5 would destabilize the interactions between the Dit-2 arm-hand extensions and the receptor-binding domains, by inserting itself between the RBP and the Dit hand loop (Figure 2C) and would yield an activated/opened conformation of the baseplate. With the aim of determining the 3D structure of the activated baseplate in the virions, we mixed p2 virions with an excess of VHH5 and imaged the sample by nsEM. While the baseplate of a large number of p2 virions remained in a rest form, a few virions surprisingly form dimers associated via their baseplates (Figure 4A). The bulky, rectangular aspect of the dimeric baseplates suggested that they might exhibit an activated conformation with the RBPs pointing downwards.

Analysis of these dimeric virions by reference-free 2D classification revealed rather large baseplates at the tail tips, as compared with the compact morphology of the baseplate rest form. Such baseplate size and form were reminiscent of the crystal structure of the p2 activated baseplate [6] (Figure 4B and Figure 1F). The final 3D reconstruction of these assemblies at 28.7 Å resolution, imposing D6 symmetry, revealed baseplates in their activated state with the receptor-binding domains pointing in the direction opposite to the capsid (Figure 4C and Appendix A). Moreover, two layers of densities at the baseplate-baseplate interface, consistent with the topology of VHH5 binding to RBPs, hold the dimeric phages particles together (Figure 4D).

We produced a topological model of the activated baseplate within the virions by rigid-body fitting structural models within the nsEM map. The Dit-1/Tal/RBP assembly from the activated baseplate crystal structure fitted well in the map (Appendix A). In order to gain insight into the molecular determinants mediating the baseplate-baseplate interface, we fitted the structure of one RBP trimer bound to three VHH5 (RBP_3_-VHH5_3_) (PDB ID 2BSE; [11]) and generated D6 symmetry-related VHH5 copies to fill in the whole baseplate-baseplate interface (Appendix A). The resulting structural model indicates that the baseplate-baseplate interface mainly relies on contacts between opposite (RBP_3_-VHH5_3_) tripods (Figure 4E).

Lastly, we fitted our model of Dit-2, obtained from the baseplate resting state, in the map and above Dit-1 (Appendix A) (Figure 5A). To note, the map volume of the arm-hand extension appears larger than that of the experimental map. We postulate that the mobility of this domain is increased once the contact with the receptor-binding domain has been released (Figure 5B), leading to a smaller map volume (Figure 5A).

### 3.3. The Tal Trimer Can Be Open or Closed in the Virion Baseplate Activated State

Our 3D reconstruction of the virion baseplate in its active form shows the open conformation of the Tal trimer at the distal tail end, as expected from the crystal structure of the p2 baseplate with the RBPs pointing downwards (Figure 1F). However, 3D classification of the dimeric baseplates allowed us to identify and distinguish a class of particles in which the Tal trimer is open (74% of selected particles) and, surprisingly, a class of particles in which the Tal trimer is closed (26% of selected particles) (Appendix A). We refined these opened and closed 3D classes, imposing D3 symmetry to be consistent with the Tal assemblies, that led to final 3D reconstructions at 35.4 Å and 42.2 Å resolution, respectively. The main variations between both 3D reconstructions are located in the Tal density, as confirmed by the difference map (Figure 6A–C). Our topological model of the activated baseplate with an opened Tal trimer (Figure 5B) fits well in the 3D reconstruction with an opened Tal (Figure 6E), while it leaves an empty density in the 3D reconstruction with the closed Tal (Appendix A). Therefore, we also produced a topological model of the activated baseplate with a closed Tal trimer including the Dit-1, Dit-2, and RBPs from the activated baseplate and the Tal trimer form the baseplate resting state (PDB ID 2WZP). This model fits well in the 3D reconstruction with a closed Tal (Figure 6D), while it is out of density in the 3D reconstruction with an opened Tal (Appendix A). The above results indicate that the RBP conformational changes from their resting to activated state is not correlated to the opening of the Tal trimer (Figure 6F).

## 4. Discussion

Comprehensive examination of nsEM images of p2 native virions failed to reveal the presence of activated baseplates, even in the presence of Ca^2+^. This suggests that this cation alone is not sufficient to trigger the baseplate conformational change, while it may stabilize the activated conformation when obtained. We first hypothesized that the VHH5 addition to p2 virions could compete for a common site between the VHH5 and the Dit-2 arm-hand extension contacting the receptor-binding domain. However, this hypothesis has to be dismissed since our baseplate in the present topological model shows that the VHH5 and Dit-2 arm-hand extension binding sites on trimeric RBP are distinct. Nevertheless, while two surface-exposed VHH5-binding sites can be accessed without impairing the baseplate resting state (Figure 2C), the third VHH5-binding site is hindered by the baseplate. The binding of this third VHH5 can occur only if “breathing” movements of the RBPs allow a VHH5 to sneak in between the RBP and the baseplate core. The high affinity between VHH5 and RBP (approx. 1 nM) would prevent further dissociation. Thus, it is tempting to speculate that a first binding event may destabilize the baseplate and trigger a cascade of VHH5 binding on the other receptor-binding domains. Once achieved, this opened baseplate conformation with the receptor-binding domains pointing in the direction opposite to the capsid, will be further stabilized by VHH5 back-to-back pairing from two complexed virions. This baseplate pairing has been previously observed when complexing the baseplate of phage TP901–1 by various VHHs [21].

But how does this opening mechanism compare with the natural activation of p2 virion baseplate? It has been reported that the conformational change and activation of myophage T4 baseplate is triggered by mechanical stress when at least three long-fibers attach to their host receptor [22]. For the siphophage p2, the baseplate resting conformation harbors two accessible saccharidic-binding sites available for cell wall polysaccharide (CWPS) binding. The concerted effort of several chains of CWPS would destabilize the baseplate enabling the interaction of further CWPS chains at the third receptor-binding site. Then, the activated conformation of the baseplate, with the RBPs pointing towards the cell surface, would be maintained because these CWPS are anchored within the cell wall, and it would also be locked by Ca^2+^-binding between the Dit-1 loops [6].

What would be the advantage of such an activation mechanism? We suggest that this activation mechanism may participate in host selection. Indeed, lactococcal phages of the Skunavirus genus are mostly specialists as they usually have a narrow host range and thrive in the dairy ecosystem. As the saccharidic receptors vary from one lactococcal strain to another, they have to strongly connect their numerous RBPs with the surface-exposed binding sites (Figure 2C) in order to make the third receptor-binding site accessible. This would require a high affinity interaction between the RBPs and their receptors to compete with the interaction between the RBP and the Dit-2 arm-hand extension. In agreement with this hypothesis, we have previously shown that while phage p2 exhibits 100% selectivity for *L. lactis* strain MG1363 over strains 3107 or SMQ-388, phage TP901–1 exhibits only 75% selectivity (Table 3 in ref. [23]).

We previously postulated that the p2 baseplate conformational rearrangement was coupled with Tal opening, followed by TMP release and DNA ejection [6]. However, we observed in this study Tal in closed or opened conformations (Figure 6). Therefore, Tal opening may not be concomitant to the baseplate activation, but rather occurs in a subsequent step. This would ensure a strong virion binding to the host cell wall prior to Tal opening and DNA ejection, thereby ensuring an efficient start to the lytic cycle.

## 5. Conclusions

We present here the activated form of the p2 virion baseplate, a conformation compatible with the activated baseplate crystal structure. Analysis of the resting and activated baseplates of the virion confirmed the presence of a second Dit hexamer, Dit-2, as a component of the baseplate. This Dit-2 hexamer is positioned back-to-back with the first Dit (Dit-1), as they attach together using their N-terminal domain ring. The Dit-2 hexamer exhibits striking conformational changes as compared to the Dit-1 hexamer. In particular, the peripheral galectin domains drift toward the baseplate central axis to form a compact ring. This ring is sandwiched between the Dit-2 N-terminal domain and the last MTP ring of the tail. The overall conformation of Dit-2 is conserved in the resting and activated states of the virion. Interestingly, the Dit arm-hand extension interacting with the RBPs exhibits important structural differences in both Dits. However, it is not possible to pinpoint the Dit-2 residues contacting the receptor-binding domain with our model. Lastly, we suggest an uncoupling between baseplate activation, characterized by a large reorientation of the RBPs, and Tal opening.

## Figures and Tables

**Figure 1 viruses-12-00878-f001:**
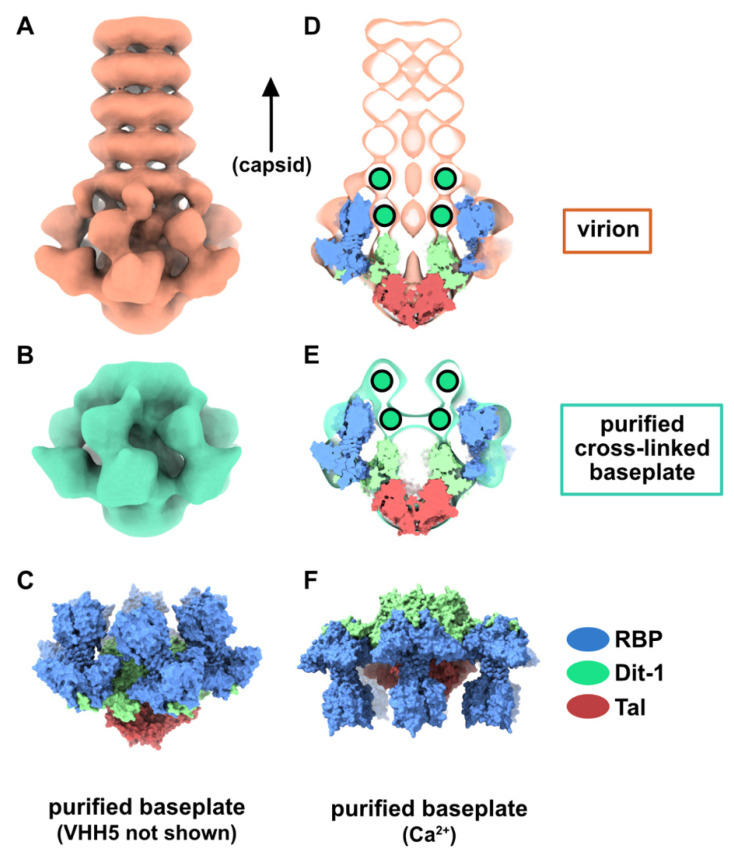
The resting and activated states of the phage p2 baseplate. (**A**) Negative-staining 3D reconstruction of phage p2 virion tail tip and baseplate (EMD-1699, [6]). The map is contoured at 2.5σ. (**B**) CryoEM 3D reconstruction phage p2 purified and cross-linked baseplate (EMD-1706 [6]). The map is contoured at 2.5σ. (**C**) Crystal structure of phage p2 purified baseplate in complex with VHH5 (PDB ID 2WZP, [6]). VHH5 molecules are not shown for clarity. (**D**,**E**) Central section of the 3D reconstructions shown in panels A and B with the baseplate crystal structure (shown in panel **C**) fitted in the maps. The green/black dots indicate unassigned EM densities/EM density rings. (**F**) The crystal structure of phage p2 purified baseplate in activated conformation, with the receptor-binding domains pointing in the opposite direction to the capsid, is shown for comparison (PDB ID 2X53 [6]).

**Figure 2 viruses-12-00878-f002:**
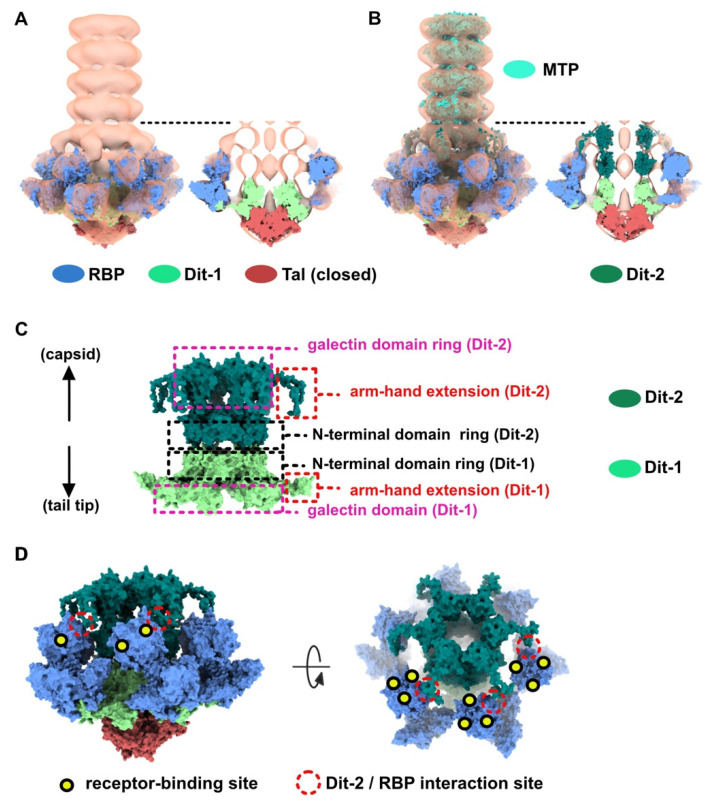
Topological model of the p2 baseplate resting state in virions. (**A**) 3D reconstruction of p2 virion’s tail tip and baseplate (EMD-1699) and surface representation of p2 receptor-binding protein (RBPs), Dit-1 and Tal crystal structures (PDB ID 2WZP) fitted in the map. The RBPs point towards the capsid and the Tal trimer is closed. An overall view (left) and a central section (right) are shown. The dotted line indicates the junction between the tail (MTPs) and the baseplate. (**B**) Same views as in A with the model of the second Dit hexamer (Dit-2) fitted in the map (surface representation). Four hexameric rings of the staphylococcal phage 80α major tail proteins (MTPs) are also shown in the tail as model structures for p2 MTPs (PDB ID 6V8I) [20]. (**C**) Topological model of the Dit-1 and Dit-2 assembly highlighting the organization of their N-terminal domain, arm-hand extension and galectin domain. (**D**) Topological model of the baseplate resting state (RBP, Tal, Dit-1, Dit-2) in p2 virions (PDB ID 6ZJJ). Side and tilted views are shown. The red dotted circles indicate the interaction between the Dit-2 arm-hand extension and one RBP. The receptor-binding sites in trimeric RBPs are indicated with yellow/black dots.

**Figure 3 viruses-12-00878-f003:**
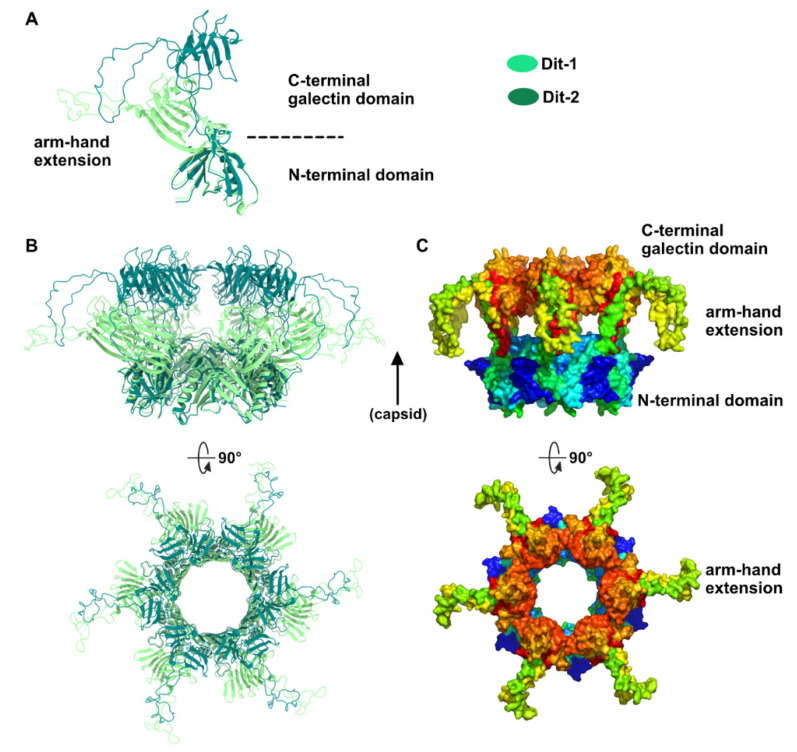
Dit structures. (**A**) Ribbon representation of superimposed Dit-1 monomer crystal structure (PDB ID 2WZP) and Dit-2 monomer model. The color code is indicated. (**B**) Ribbon representation of superimposed Dit-1 hexamer crystal structure and Dit-2 hexamer model (orthogonal views). (**C**) Surface representation of the hexameric Dit-2 model (orthogonal views). The rainbow coloring mode highlights the Dit-2 N-terminal bottom ring (blue) and the C-terminal galectin top ring (red) including the arm-hand extension (green and yellow).

**Figure 4 viruses-12-00878-f004:**
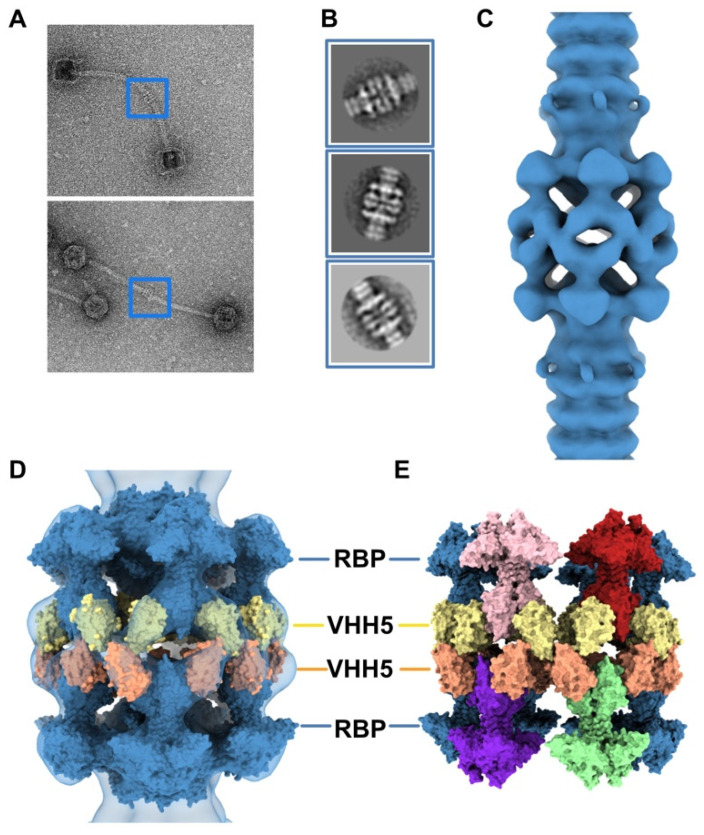
3D reconstruction of activated baseplate in VHH5-bound p2 virions. (**A**) Representative phage p2 virions in complex with VHH5 (virion-virion assembly) observed in micrographs. Baseplate-baseplate assemblies (blue boxes) were selected for image processing. (**B**) Representative 2D classes of baseplate-baseplate assemblies. (**C**) 3D reconstruction of p2 tail tip and baseplate complexed with VHH5 and forming a dimer (EMD-11226). The map is contoured at 2.5σ. (**D**) Surface representation of the p2 baseplate in its activated conformation (RBPs pointing in the direction opposite to the capsid) (PDB ID 2X53, blue), fitted in the map. Surfaces of VHH5 (PDB ID 2BSE [11]) in contact with the upper RBPs (yellow) and lower RBPs (orange) are also shown. (**E**) Surface representation of the VHH5-bound RBPs from the upper and lower baseplates as shown in D. Note that VHH5 in the upper baseplate (yellow) interact with VHH5 in the lower baseplate (orange).

**Figure 5 viruses-12-00878-f005:**
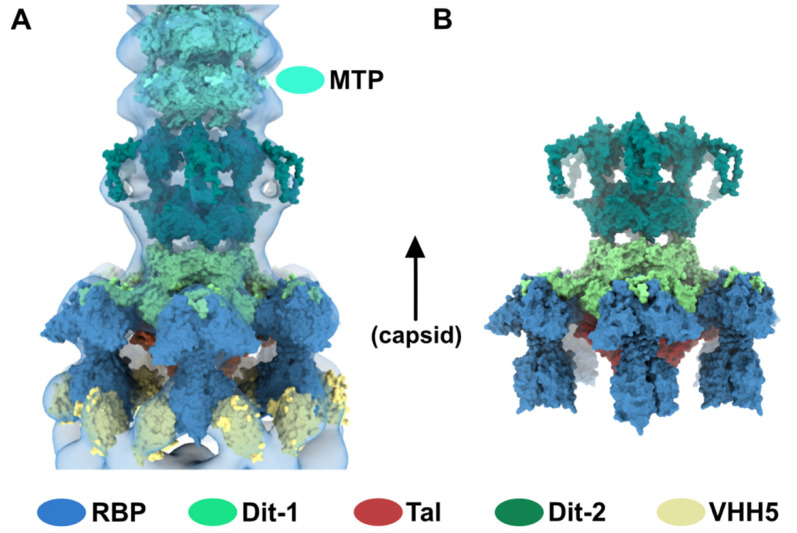
Topological model of the activated baseplate in p2 virions. (**A**) Surface representation of the baseplate components (PDB ID 2X53) and VHH5 (PDB ID 2BSE) fitted in the 3D reconstruction of VHH5-bound p2 virions (close-up on one virion). Rings of phage 80α MTPs are also shown in the tail as models of p2 MTPs. The map is contoured at 2.5σ. The color code is indicated. (**B**) Topological model of the activated baseplate (RBP, Tal, Dit-1, Dit-2) in p2 virions (PDB ID 6ZIH).

**Figure 6 viruses-12-00878-f006:**
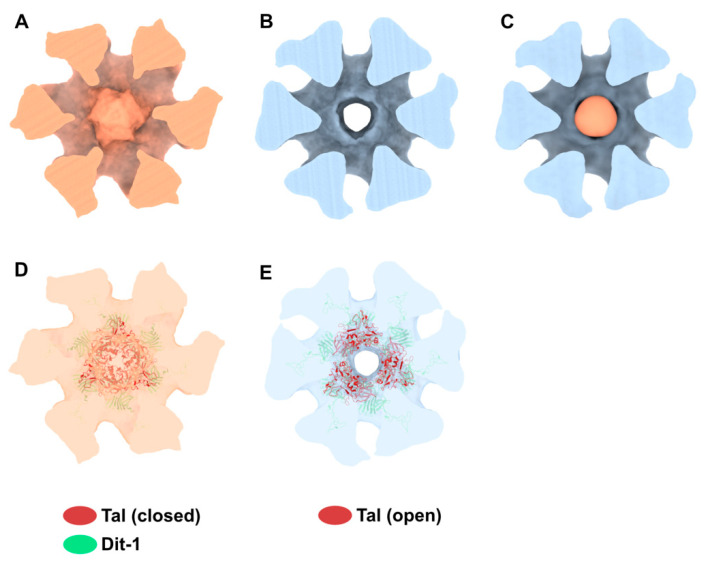
Activated baseplate in p2 virions with open and closed Tal. (**A**,**B**). 3D reconstructions of the baseplate in p2 virions, with Tal in closed (panel A, EMD-11225) and open (panel B, EMD-11224) conformations. Views along the tail axis towards the capsid. (**C**) The difference map calculated by subtracting the map shown in panel B (open Tal) to the map shown in panel A (closed Tal) is colored in light salmon. All three maps are contoured at 6σ. (**D**,**E**) Ribbon representation of the Tal trimer in closed (PDB ID 2WZP) and open (PDB ID 2X53) conformation fitted into the nsEM maps. The hexameric Dit-1 is also shown. The RBPs and Dit-2 are not shown for clarity. Maps are contoured at 5σ. The color code is indicated.

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
