# Peer review of "Structural Insights into Lactococcal Siphophage p2 Baseplate Activation Mechanism"

_viruses, 2020, doi:10.3390/v12080878_

Round 1

Reviewer 1 Report

The study of S. Spinelli et al. is an excellent work in which the authors focused on the analysis of the activated structure of the p2 virion baseplate. The paper is well written. Well-designed research and well-presented work. Authors have identified the current research gap related to the topic and justified the research question well. Research objectives are logical and clearly stated. Methods are clearly explained and fully justified with relevant references. Use of figures and table for results presentation are appropriate, well-designed and clear and presented in a concise manner. Clear, correct and insightful discussion of the results obtained in the context of the current study is evident and the major conclusions supported the key arguments. Additionally, It should also be emphasised that the authors are outstanding experts in this field, therefore I suggest publishing this manuscript in its present form.

Author Response

reviewer 1

The study of S. Spinelli et al. is an excellent work in which the authors focused on the analysis of the activated structure of the p2 virion baseplate. The paper is well written. Well-designed research and well-presented work. Authors have identified the current research gap related to the topic and justified the research question well. Research objectives are logical and clearly stated. Methods are clearly explained and fully justified with relevant references. Use of figures and table for results presentation are appropriate, well-designed and clear and presented in a concise manner. Clear, correct and insightful discussion of the results obtained in the context of the current study is evident and the major conclusions supported the key arguments. Additionally, It should also be emphasised that the authors are outstanding experts in this field, therefore I suggest publishing this manuscript in its present form.

Our answer

We sincerely thank the reviewer for the consideration and positive appreciation of our manuscript. We appreciate the acknowledgement of our effort to present a clear, argued and insightful work.

Reviewer 2 Report

The paper from Spinelli et al reports on the negative stain, low resolution structure of the baseplate of the p2 virion and of a VHH-activated state. Based on the fitting of a previously obtained crystal structure of the baseplate alone in both conformations, and on the conformational heterogeneity of the activated baseplate in the virion, the authors conclude in an uncoupling between baseplate activation and opening. The authors also confirm their previous proposition that a second ring of one of the baseplate component would fit between the baseplate and the tube of the major tail protein.

            The experiments are well planned, performed carefully and well illustrated. The authors cite the literature adequately, and give credit to related research. Some of the conclusion reached are however extremely surprising, and should be discussed further:

Major points:

In the case of the uncoupling between baseplate activation and opening, as the activation is triggered artificially by VHH binding and not by receptor binding, the “uncoupling” as discussed is convincing. After “activation”, two conformations of the baseplate are observed, both with the Receptor Binding Proteins (RBP) pointing downwards, but with the Tal protein either closed or open to allow the DNA out. In that latter conformation, it is extremely surprising that the DNA, or at least the Tape Measure Protein (TMP) should not have been expelled. I have tried to carefully look at the provided negative stained images to see hints in that direction. However, the image of figure 4 and that of Figure S1 are the same, severely limiting the possible statistics! It seems that in the upper boxed baseplate, the tails are still full of the TMP, whereas in the lower one, tails are empty. Does that relate to the Tal being closed and empty, respectively, for these particles? How do the authors explain that the capsids are full in virions in which the Tal is  open? This should absolutely be discussed.

An even more surprising conclusion of the paper is the presence of a second ring of the Dit protein located between the baseplate and the tail tube. If I understand correctly (it is not stated/described clearly in the manuscript), this Dit 2 protein would account for two rings of the tube: one formed by the “hexamerisation” domain of Dit, which in Dit 1 sit above the Tal trimer, and the other ring would be formed by the galectin domain. This galectin ring would sit between the ring formed by the “hexamerisation” domain and the first (or last) Major Tail Protein (MTP) of the main tube. This would be extremely surprising. Have galectin domains already been shown to form oligomers/hexamers in other studies? The duplication of the Dit ring is already quite unusual: in other phages, either sipho-myo or other bacterial contractile machineries, there is direct interaction of the MTP with the Dit protein (or even no Dit as in GTA!), or, in more sophisticated baseplates, there are one or several rings of “baseplate MTP”.  Given the extremely conserved architecture of the tail of sipho and myophages, the suggestion of a ring of a galectin domain is extremely surprising. Given that, in addition, this result is based on low resolution, negative stain data, the authors should be extremely cautious, and explore other possibilities to interpret the missing densities.  Furthermore, if I remember well, this idea of a second Dit ring has already been suggested by the authors but with a back-to-back conformation of the two Dit rings (Sciara et al). It is not clear whether the authors still favour this configuration. This should also be discussed.

Minor points:

- p2, line62, the Tal protein is also more commonly called Hub protein in other phages. It could help to use that name as well, to help with the reader that are not familiar with the lactophages.

- figure 1 is not detailed enough in the organisation of the Dit 1 hexamer, necessary to understand how the Dit 2 ring would arrange.

- p5, last paragraph. A figure illustrating the fit of the second Dit should be presented, as it is not clear at all how the authors arrange that protein in the EM density. Figure 2 is not very informative.

- p6, last paragraph. A figure as suggested above would prevent having to detail the location of short stretches of residues in a 30-Å negative stain map… Strange to be so detailed about those loops, and so vague about the organisation of the entire domain (I still have not understood how they were organised: back-to-back or in the same orientation…).

- p7, lines 210-214. How VHH would activate the baseplate is not clear, please rephrase.

- p7, line217-218. “Close inspection of individual images of the dimeric baseplates suggested that they might exhibit an activated conformation.” Please say what makes you say that. Orientation of the RBP? Aspect of the tail (absence of the TMP)? Please elaborate.

- p8, line 225. “consistent with the VHH5 binding mode" What is a VHH binding mode?

- Figure 4-S1: Please provide at least two different images of the VHH-activated virions!!

- Figure 4-5: the choice of colours does not seems optimal. Blue EM density and blue model… In figure 5, the different shades of blue/blue green/green are not very distinguishable either…

- p8, line 254, define MTP.

- Figure 5B, repeated in Figure 6F, does not seem very useful in either case.

- Figure 6D-E. The Dit 1 hexamer in green do not shown well at all (I missed them on the print).

- p11, lines 332-333. “as they attach together using their N-terminal belt ring.” Please define N-terminal belt ring.

Author Response

reviewer 2

The paper from Spinelli et al reports on the negative stain, low resolution structure of the baseplate of the p2 virion and of a VHH-activated state. Based on the fitting of a previously obtained crystal structure of the baseplate alone in both conformations, and on the conformational heterogeneity of the activated baseplate in the virion, the authors conclude in an uncoupling between baseplate activation and opening. The authors also confirm their previous proposition that a second ring of one of the baseplate component would fit between the baseplate and the tube of the major tail protein.

The experiments are well planned, performed carefully and well illustrated. The authors cite the literature adequately, and give credit to related research. Some of the conclusion reached are however extremely surprising, and should be discussed further:

Our answer

We sincerely thank the reviewer for the consideration and helpful evaluation of our manuscript. We appreciate the accurate summary of our work and the acknowledgment of our effort to present a clear and well-illustrated work.

We have addressed each of the points raised by the reviewer and revised the manuscript accordingly, which is improved as a result.

Major points:

  • In the case of the uncoupling between baseplate activation and opening, as the activation is triggered artificially by VHH binding and not by receptor binding, the “uncoupling” as discussed is convincing. After “activation”, two conformations of the baseplate are observed, both with the Receptor Binding Proteins (RBP) pointing downwards, but with the Tal protein either closed or open to allow the DNA out. In that latter conformation, it is extremely surprising that the DNA, or at least the Tape Measure Protein (TMP) should not have been expelled. I have tried to carefully look at the provided negative stained images to see hints in that direction. However, the image of figure 4 and that of Figure S1 are the same, severely limiting the possible statistics! It seems that in the upper boxed baseplate, the tails are still full of the TMP, whereas in the lower one, tails are empty. Does that relate to the Tal being closed and empty, respectively, for these particles? How do the authors explain that the capsids are full in virions in which the Tal is  open? This should absolutely be discussed.

Our answer

We acknowledge the reviewer’s concern on the relationship between Tal opening and DNA ejection. We agree that showing only viruses with full capsids in Figure 4A and Figure S1 is misleading. In our dataset, we do observe dimers of virion with empty capsid, which are actually the majority, as well as dimers of virions with full capsids (see some micrographs below).

Representative electron micrographs showing virions with empty (top row) or full (bottom row) capsids.

In these images, the tails of virions with capsid free of DNA look empty while the tails of virions with DNA-containing capsid look filled by the TMP.

We have amended the manuscript for a better understanding. In particular, our revised Figure 4A shows micrographs of dimeric virions with empty and full capsids. Different micrographs of dimeric virions with empty and full capsids are also shown in Figure S1A. The legends were modified accordingly (p8, line 232 ‘Representative phage p2 virions in complex with VHH5 (virion-virion assembly) observed in micrographs.’ and supplementary ‘Figures S1. Negative-staining 3D reconstructions of p2 baseplate in virions complexed to VHH5. A. Representative p2 virions mixed with VHH5 observed in micrographs.’).

  • An even more surprising conclusion of the paper is the presence of a second ring of the Dit protein located between the baseplate and the tail tube. If I understand correctly (it is not stated/described clearly in the manuscript), this Dit 2 protein would account for two rings of the tube: one formed by the “hexamerisation” domain of Dit, which in Dit 1 sit above the Tal trimer, and the other ring would be formed by the galectin domain. This galectin ring would sit between the ring formed by the “hexamerisation” domain and the first (or last) Major Tail Protein (MTP) of the main tube. This would be extremely surprising. Have galectin domains already been shown to form oligomers/hexamers in other studies? The duplication of the Dit ring is already quite unusual: in other phages, either sipho-myo or other bacterial contractile machineries, there is direct interaction of the MTP with the Dit protein (or even no Dit as in GTA!), or, in more sophisticated baseplates, there are one or several rings of “baseplate MTP”.  Given the extremely conserved architecture of the tail of sipho and myophages, the suggestion of a ring of a galectin domain is extremely surprising. Given that, in addition, this result is based on low resolution, negative stain data, the authors should be extremely cautious, and explore other possibilities to interpret the missing densities.  Furthermore, if I remember well, this idea of a second Dit ring has already been suggested by the authors but with a back-to-back conformation of the two Dit rings (Sciara et al). It is not clear whether the authors still favour this configuration. This should also be discussed.

Our answer

  • The key point in identifying the second Dit domain has been provided in our PNAS (210) and is repeated here, as seen ib Figure 1E. In the expressed and cross linked baseplate, after fitting the three proteins (18 RBPs, 6 Dits, 3 Tals), two rings of density remained not satisfied above the Dit. We could satisfy these densities with a second Dit, as no other protein was available. The lower ring could accomodate the N-terminal domains of the second Dit hexamer, while there was only the galectin domains availabe for the second ring. In PBAS (2010) and here, in more details, we fitted the Galectins in order to assemble as a compact ring that satify the EMmap. Nicely enough, the arm-and-hand loops of the galectins fit well, although without atomic detail, in the map.  
  • As we show in our recent review in Viruses (Goulet, A.; Spinelli, S.; Mahony, J.; Cambillau, C., Conserved and Diverse Traits of Adhesion Devices from Siphoviridae Recognizing Proteinaceous or Saccharidic Receptors. Viruses 2020, 12, (5). ), Dits are well conserved in Siphoviridae and most of the time form an unique hexameric ring using their N-terminal domains, with an extra domain at the periphery (Galectin or OB)-fold, or no extra domain (T5)). The only exceptions to date are phages p2and related (Skunaphages) and 1358 who exhibit an activation mechanism. To keep their baseplate in rest mode, these phages use a second Dit hexamer of which the galectins arm-and-hand loops lock the RBPs.

Minor points:

- p2, line62, the Tal protein is also more commonly called Hub protein in other phages. It could help to use that name as well, to help with the reader that are not familiar with the lactophages.

Our answer

The use of "Hub" is frequent in Myoviridae, not in Siphoviridae where it extends beyonf loctococcal phages as shown in recent exemples of listeria and staphylococcal phages:Dunne, M., Rupf, B., Tala, M., Qabrati, X., Ernst, P., Shen, Y., et al. (2019) Reprogramming Bacteriophage Host Range through Structure-Guided Design of Chimeric Receptor Binding Proteins, Cell Rep 29: 1336-1350 e1334. and Kizziah, J.L., Manning, K.A., Dearborn,A.D., Dokland, T. , Structure of the host cell recognition and penetration machinery of a Staphylococcus aureus bacteriophage. PLOS-Pathogens 2020 16:e1008314. See also our review in Viruses: Goulet, A.; Spinelli, S.; Mahony, J.; Cambillau, C., Conserved and Diverse Traits of Adhesion Devices from Siphoviridae Recognizing Proteinaceous or Saccharidic Receptors. Viruses 2020, 12, (5).

- figure 1 is not detailed enough in the organisation of the Dit 1 hexamer, necessary to understand how the Dit 2 ring would arrange.

and

- p5, last paragraph. A figure illustrating the fit of the second Dit should be presented, as it is not clear at all how the authors arrange that protein in the EM density. Figure 2 is not very informative.

Our answer

A more detailed view of the Dit1/Dit2 arrangement is provided now in Figuure 2 C

- p6, last paragraph. A figure as suggested above would prevent having to detail the location of short stretches of residues in a 30-Å negative stain map… Strange to be so detailed about those loops, and so vague about the organisation of the entire domain (I still have not understood how they were organised: back-to-back or in the same orientation…).

Our answer

Concerning the Dit1/Dit2 arrangement, it is indicated in the text line 77: "Therefore, we proposed that one of the rings could be formed by the a Dit N-terminal domain back to back with the first Dit t N-terminal domain (referred to as Dit-1) hexameric ring, while the second ring (referred to as Dit-2) could be formed by the Dit C-terminal galectin domains, rearranged to form a hexamer [6]. " To note, this is exactly wht was stated in the p2 baseplate PNAS paper (2020).

We explicitely stated in the text that the loop topology is tentative and only compatible with the size of the EM map: Therefore, we also modeled their backbone to fit them into the map, although their exact position could not be acertained.

- p7, lines 210-214. How VHH would activate the baseplate is not clear, please rephrase.

Our answer

We completed the sentence as indicated below: "binding of VHH5 would destabilize the interactions between the Dit-2 arm-hand extensions and the receptor-binding domains, by insering itself between the RBP and the Dit hand loop"

- p7, line217-218. “Close inspection of individual images of the dimeric baseplates suggested that they might exhibit an activated conformation.” Please say what makes you say that. Orientation of the RBP? Aspect of the tail (absence of the TMP)? Please elaborate.

Our answer

The bulky, rectangular aspect of the dimeric baseplates, as compared to the conical shape of the baseplate rest form with the RBP pointing towards the capsid, made us think that they might represent activated conformation with the RBP pointing downwards.

We have amended this sentence to be more precise:

P7, lines xxx: “The bulky, rectangular aspect of the dimeric baseplates suggested that they might exhibit an activated conformation with the RBPs pointing downwards.”

- p8, line 225. “consistent with the VHH5 binding mode" What is a VHH binding mode?

Our answer

Changed to " consistent with the topology of the VHH5 binding to RBPs"

- Figure 4-S1: Please provide at least two different images of the VHH-activated virions!!

Our answer

Provided in the new Supplementary Figure 1

- Figure 4-5: the choice of colours does not seems optimal. Blue EM density and blue model… In figure 5, the different shades of blue/blue green/green are not very distinguishable either…

Our answer

We extensively used different coloring modes. we noticed that when map and models have different colors, the result is very confusing.

- p8, line 254, define MTP.

Our answer

Provided: Rings of phage 80α Major Tail Proteins (MTPs) are also shown

- Figure 5B, repeated in Figure 6F, does not seem very useful in either case.

Our answer

Figure 6F removed. We think that Figure 5B is a good complement of Figure 2 C.

- Figure 6D-E. The Dit 1 hexamer in green do not shown well at all (I missed them on the print).

Our answer

Printing very often results in very poor colors. We chek on the screen and the result seemed rather clear.

- p11, lines 332-333. “as they attach together using their N-terminal belt ring.” Please define N-terminal belt ring.

Our answer

The term "belt" hs been replaced by "domain", used in the manuscript

Reviewer 3 Report

This manuscript reported of an interesting and specific study of the phage baseplate activation mechanism.

I have no comments about it due also to the highly specific argument that is reported in this study.

Author Response

reviewer 3

This manuscript reported of an interesting and specific study of the phage baseplate activation mechanism.

I have no comments about it due also to the highly specific argument that is reported in this study.

Our Answer

We sincerely thank the reviewer for the consideration and positive appreciation of our manuscript. We appreciate the acknowledgement of our effort to present a clear, argued and insightful work.

Reviewer 4 Report

In my view, this is an excellent work. The manuscript is well-written, and the research question is interesting and timely. The findings will contribute to the scientific advancements in this field. Few minor suggestions to further improve the quality of this manuscript is given below;

Lines 37, 41 and 111: Escherichia coli and Lactococcus lactis: Scientific names should be in Italic. Please correct and check throughout the manuscript.  

Line 45: several groups – can you please state few examples from these several groups in here ?

Section 2. Materials and Methods: It is better to provide the full details of material providers. For example, in line 117, you mentioned as “ Agar Scientific”. Better to provide the city/ state and the country of this provider as well. Please consider revising throughout the manuscript.

Author Response

reviewer 4

In my view, this is an excellent work. The manuscript is well-written, and the research question is interesting and timely. The findings will contribute to the scientific advancements in this field. Few minor suggestions to further improve the quality of this manuscript is given below;

Lines 37, 41 and 111: Escherichia coli and Lactococcus lactis: Scientific names should be in Italic. Please correct and check throughout the manuscript.  

Our answer

Corrected

Line 45: several groups – can you please state few examples from these several groups in here ?

Our answer

"into several groups (e.g. Skunavirus, P335, etc)"

Section 2. Materials and Methods: It is better to provide the full details of material providers. For example, in line 117, you mentioned as “ Agar Scientific”. Better to provide the city/ state and the country of this provider as well. Please consider revising throughout the manuscript.

Our answer

Corrected

Round 2

Reviewer 2 Report

The authors have improved the paper.

It is still not clear how a galectin domain could form a ring inserting itself in the tube of the phage. Furthermore, in figure 3, in the model with the Dit2, the tube does not seem to be sealed, i.e. the N-terminal domain and the galectin domain do not seem to interact. This is extremely strange for a phage tail tube. The authors should discuss that point al least.

Define 80alphaMTP in the legent of figure 2 rather than in the legend of figure5.

Author Response

- It is still not clear how a galectin domain could form a ring inserting itself in the tube of the phage.

OUR ANSWER

This is an experimental observation. The mechanism by which this occurs remains to be determined.

- Furthermore, in figure 3, in the model with the Dit2, the tube does not seem to be sealed, i.e. the N-terminal domain and the galectin domain do not seem to interact. This is extremely strange for a phage tail tube. The authors should discuss that point al least.

OUR ANSWER

Why the tube should be "sealed"? It is enough that the rings are maintained together. A space is always observed between the packed rings, within the linker region. There are several exemples, such as that of the MTPs of phage80alpha (see the joint figure in the pdf file). In our case, the linker between the N-terminal ring and the galectin ring does not fill completely the space between monomers, but the link being made of covalent bonds, it is strong enough to obtain a stable tube.

- Define 80alphaMTP in the legent of figure 2 rather than in the legend of figure5.

OUR ANSWER

Corrected
